# High prevalence of Zika virus infection in populations of *Aedes aegypti* from South-western Ecuador

**Andrea López-Rosero[1], Rachel Sippy[2¤a], Anna M. Stewart-Ibarra[2¤b], Sadie J. Ryan[3], Erin Mordecai[4], Froilán Heras[2], Efraín Beltrán[5], Jaime A. Costales[1], Marco Neira[1¤c]***

**1** Centro de Investigación para la Salud en América Latina (CISeAL), Facultad de Ciencias Exactas y Naturales, Pontificia Universidad Católica del Ecuador, Quito, Ecuador, **2** Institute for Global Health and Translational Science and Department of Medicine, SUNY Upstate Medical University, Syracuse, New York, United States of America, **3** Quantitative Disease Ecology and Conservation (QDEC) Laboratory, Department of Geography and Emerging Pathogens Institute, University of Florida, Gainesville, Florida, United States of America, **4** Department of Biology, Stanford University, Stanford, California, United States of America, **5** Universidad Técnica de Machala, Machala, El Oro, Ecuador

¤a Current address: School of Mathematics and Statistics, University of St. Andrews, Fife, United Kingdom
¤b Current address: Inter-American Institute for Global Change Research, Montevideo, Uruguay
¤c Current address: Climate and Atmosphere Research Center (CARE-C), The Cyprus Institute, Nicosia, Cyprus
* m.neira@cyi.ac.cy

**Data Availability Statement:** All relevant data are within the paper and its Supporting Information files.

## Abstract

We performed an arboviral survey in mosquitoes from four endemic Ecuadorian cities (Huaquillas, Machala, Portovelo and Zaruma) during the epidemic period 2016–2018. Collections were performed during the pre-rainy season (2016), peak transmission season (2017) and post-rainy season (2018). *Ae. aegypti* mosquitoes were pooled by date, location and sex. Pools were screened by RT-PCR for the presence of ZIKV RNA, and infection rates (IRs) per 1,000 specimens were calculated. A total of 2,592 pools (comprising 6,197 mosquitoes) were screened. Our results reveal high IRs in all cities and periods sampled. Overall IRs among female mosquitoes were highest in Machala (89.2), followed by Portovelo (66.4), Zaruma (47.4) and Huaquillas (41.9). Among male mosquitoes, overall IRs were highest in Machala (35.6), followed by Portovelo (33.1), Huaquillas (31.9) and Zaruma (27.9), suggesting that alternative transmission routes (vertical/venereal) can play important roles for ZIKV maintenance in the vector population of these areas. Additionally, we propose that the stabilization of ZIKV vertical transmission in the vector population could help explain the presence of high IRs in field-caught mosquitoes during inter-epidemic periods.

## Author summary

Zika virus (ZIKV) is a mosquito-borne virus native to Africa that can cause a wide range of symptoms in humans, including neurological complications and fetal abnormalities. ZIKV arrived to South America in 2015, and quickly spread through the continent. In Ecuador, this virus caused several thousand infections during 2016 and 2017, but its

**Funding:** RS, AMSI, SJR, MN received funding from the National Science Foundation Zika Rapid (DEB-161145). ALR, RS, AMSI, SJR, EM, FH, MN received funding from the National Science Foundation Ecology and Evolution of Infectious Diseases grant (DEB-52 1518681). EM received support from the U.S. National Science Foundation (DEB-2011147, with the Fogarty International Center), the National Institutes of Health (R35GM133439), and the Stanford University Woods Institute for the Environment and Center for Innovation in Global Health. The funders had no role in study design, data collection and analysis, decision to publish, or preparation of the manuscript.

**Competing interests:** The authors have declared that no competing interests exist.

incidence dropped suddenly thereafter, with just 11 cases reported between 2018 and 2022. The reasons behind this abrupt decrease are not well understood.

In this work, we performed an arboviral survey in mosquitoes from four endemic Ecuadorian cities during the period 2016–2018. We screened 2,592 pools (comprising 6,197 specimens) for the presence of ZIKV, and estimated infection rates in female and male mosquitoes. We found high infection rates in all cities and periods sampled. Considering the low incidence of human cases during 2018, our results suggest that in our study areas the virus could be maintained by mechanisms that either do not depend on human infection, or cause low rates of symptomatic infections. High rates of ZIKV infection in male mosquitoes found during our study suggest that vertical and/or venereal transmission routes could play important roles in the inter-epidemic maintenance of this pathogen in our study area.

## Introduction

Arthropod-borne viruses, also known as arboviruses, are usually transmitted between hosts by the bite of blood-feeding arthropods, including mosquitoes, sandflies and ticks [1]. A subgroup of arboviruses, which include dengue virus (DENV), Zika virus (ZIKV) and chikungunya virus (CHIKV), are preferentially transmitted by *Aedes aegypti* and *Ae. albopictus* mosquitoes. Because these mosquito species are highly synanthropic, *Aedes*-borne viruses are spreading quickly and have become major threats to global public health [2].

ZIKV is a member of the *Orthoflavivirus* genus, which belongs to the *Flaviviridae* family of viruses. Virions are 40–60 nm in diameter and contain a single-stranded 11-kb RNA genome, which encodes three structural and seven non-structural proteins [3,4]. Under sylvatic conditions, ZIKV circulates between non-human primates and various species of *Aedes* mosquitoes. However, humans exposed to sylvatic environments can become infected and carry the virus into urban areas, where epidemic cycles can then be maintained between susceptible humans and synanthropic *Aedes* species, principally *Ae. aegypti* and *Ae. albopictus* [4].

ZIKV infection in humans is estimated to be asymptomatic in 80% of cases [5]. When present, symptoms might include rash, low-grade fever, joint and muscle pain, periarticular edema and conjunctivitis [6,7]. A small fraction of infected individuals develops neurological complications, including Guillain-Barré syndrome, meningoencephalitis and myelitis [8,9]. ZIKV infection in pregnant women has been linked to serious complications including fetal loss, congenital microcephaly, ventriculomegalia and eye abnormalities, among others [10,11].

First isolated in Uganda in 1947, ZIKV was first reported have spread outside of Africa in the 1960s, when it was detected in Asia. For the next two decades, sporadic cases of ZIKV infection were reported within Africa and Asia, until 2007, when the virus caused an outbreak in Micronesia. In 2013 and 2014 the virus spread to French Polynesia and Easter Island, reaching Brazil in 2015 [7,12]. Within a year, ZIKV caused a massive outbreak that spread throughout the Americas, leading to the World Health Organization to declare this disease a public health emergency of international concern in 2016 [13]. In 2017, the first cases of ZIKV were reported in India [4].

With cases being widely under-reported and the dynamics of ZIKV transmission still unclear, it is critical to measure virus circulation in mosquito vector populations over time, to characterize transmission risk and modes. In this study, we aim to characterize ZIKV infection rates in vectors from Ecuador during the epidemic period (2016–2018) to understand the extent of transmission risk and the frequency of vertical transmission (via viral detection in

**Table 1. Number of dengue and Zika cases in Ecuador, 2015–2022.** Data sources:[14,15].

|  | 2015 | 2016 | 2017 | 2018 | 2019 | 2020 | 2021 | 2022 |
|---|---|---|---|---|---|---|---|---|
| Dengue | 42,473 | 14,159 | 11,387 | 3,099 | 8,416 | 19,950 | 20,592 | 16,017 |
| Zika | 1 | 2,902 | 2,331 | 9 | 0 | 0 | 2 | 0 |

non-biting male mosquitoes). Zika dynamics followed a sharp epidemic curve in Ecuador (Table 1). After appearing in Ecuador in 2015, Zika cases increased rapidly, with 2,947 and 2,423 cases reported during 2016 and 2017, respectively (Table 1). However, incidence dropped suddenly to just 9 cases in 2018, and only two new cases have been reported since that year [14] (Table 1). In this study, we screened male and female *Ae. aegypti* mosquitoes collected in the southern province of El Oro for the presence of Zika virus. We calculate infection rates for each sex, city and time period, and discuss potential biological and epidemiological implications of our data.

## Methods

### Ethics statement

Before the initiation of field activities, the protocols used in this study were dutifully approved by Institutional Review Boards in Ecuador (Luis Vernaza Hospital and Ministry of Health) and the USA (State University of New York Upstate Medical University, IRBNET ID 4177710–25). Written informed consent was obtained from all adult participants (18 years of age or older).

### Study area

Samples were collected in four cities (Machala, Huaquillas, Portovelo and Zaruma) of El Oro province, which is located in southwestern Ecuador, and borders Peru (Fig 1 and Table 2). Although all four cities present endemic transmission of *Aedes*-borne arboviruses (specifically, dengue virus), they differ in the incidence of arboviral disease: while Machala and Huaquillas present high numbers of cases every year, higher-elevation Portovelo and Zaruma present more limited incidence [16].

### Sampling period

The study was conducted between 2016 and 2018, across different seasons in different years to capture a fuller range of conditions. Peak mosquito abundance and arbovirus transmission occurs in the area during the rainy season, which takes place from December to May, and transmission persists at low levels during the rest of the year [18]. During 2016, the collection of mosquitoes was performed between July and December, corresponding to the pre-rainy season. In 2017, collections took place between January and May, which corresponds to the peak transmission season and, in 2018, collections were performed between April and August, corresponding to the post-rainy period.

### Mosquito collection, pooling and RNA extraction

Each city was geographically divided in 10 clusters. In the center of each cluster, one house was selected for participation, and additional houses (up to 5 per cluster, within a distance of 250 meters from the central house) were randomly enrolled into the study. Houses with air conditioning were excluded. Participating houses were georeferenced and issued a cluster and house identification code. Mosquito collections were performed every two weeks in these houses,

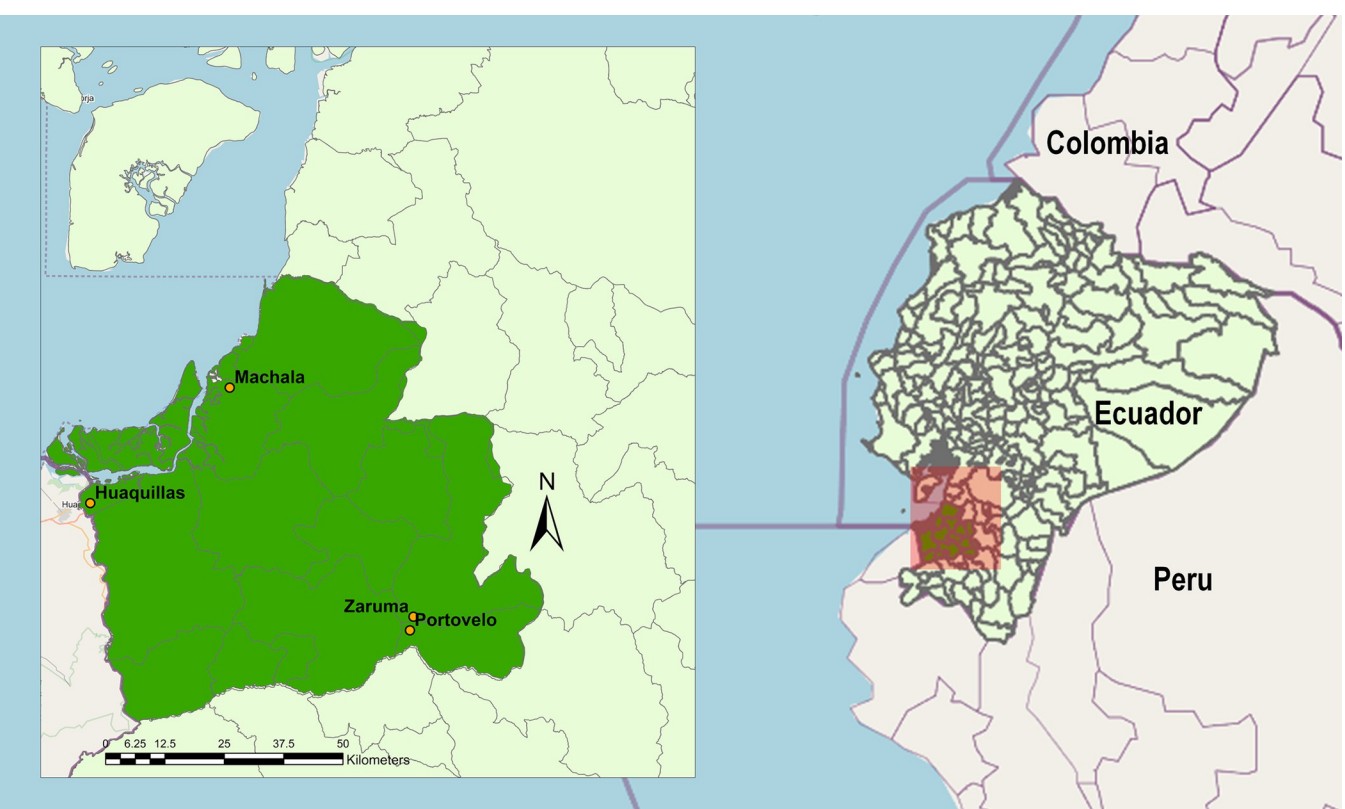

**Fig 1. Study area.** The country map on the right shows the location of El Oro province (dark green area) within Ecuador (light green area). Inset on the left corresponds to the area shaded in pink on the country map. The cities where this study took place are marked with orange dots and labelled on the inset. Areas colored in grey correspond to neighboring countries. Basemap source: OpenStreetMap (https://openstreetmap.org). Administrative boundaries shapefile source: The Humanitarian Data Exchange (https://data.humdata.org/dataset/cod-ab-ecu).

including all indoor spaces as well as porch/patio spaces. If residents were not home during a given collection period, the house was skipped for the collection period. Mosquitoes were collected using manual Prokopack aspirators [19]. Following each collection, mosquitoes were transported on ice to a field laboratory where *Ae. aegypti* specimens were identified and pooled by sex and cluster in groups of ≤10 specimens per house for further processing.

For homogenization, each pool was placed in a 2 mL sterile microcentrifuge tube containing 1mL TRI Reagent (Molecular Research Center, Inc., Cincinnati, OH, USA) and two small, sterile metal beads. Tubes were capped tightly and vortexed until no whole abdomens were visible. The homogenate was transferred to a new sterile microcentrifuge tube and stored at -20°C until transported to the laboratories at the 'Centro de Investigación para la Salud en América Latina' (CISeAL), in Quito, where samples were further processed.

**Table 2. Characteristics of cities where collections were performed.** Population data rounded from [17]. Arboviral transmission data from [16].

| City | Location | Altitude | Population | Arboviral transmission | Description |
|---|---|---|---|---|---|
| *Machala* | 3°15'09"S, 79°57'20"W | 6m | ~289,000 | High | Coastal city. Province capital with intense agricultural and commercial activity. |
| *Huaquillas* | 3°28'33"S, 80°13'33"W | 15m | ~60,000 | High | Coastal city. Borders Peru. Intensive cross-border commerce and trade. |
| *Portovelo* | 3°42'58"S, 79°37'08"W | 645m | ~14,000 | Limited | Inland. Intense mining activity. |
| *Zaruma* | 3°41'31"S, 79°36'47"W | 1,155m | ~26,000 | Limited | Inland. Intense mining activity. |

Total RNA was extracted following standard TRI Reagent manufacturer's recommendations. RNA pellets were washed in 75% EtOH, re-suspended in 50 µl of nuclease-free water and stored at -80˚C until further processing.

## cDNA synthesis

cDNA synthesis was performed using the GoScript Reverse Transcription Kit (Promega Corp., Wisconsin, USA), following manufacturer's recommendations. Briefly, 2 µl total RNA were mixed with 1 µL random primers (0.5µg/µL) and 2 µl nuclease-free water. The mixture was heated to 70˚C for 5 minutes to linearize RNA and then cooled to 4˚C for 5 minutes. Linearized RNA was then mixed with 15 µl of reverse transcription mix (containing 4 µl 5X GoScript buffer, 2 µl $MgCl_2$ 25nM, 1 µl 10 mM dNTPs and 1 µl GoScript reverse transcriptase). This cocktail was subsequently incubated at 25˚C for 5 minutes, 42˚C for 1 hour and 70˚C for 15 min. Synthetized cDNA was stored at -80˚C until further use.

## RNA extraction / cDNA synthesis control

In order to verify the success of the RNA extraction and cDNA synthesis, each sample was screened for the presence of mRNA corresponding to the housekeeping gene *Actin-1*, following the protocol described by [20]. Only samples for which *Actin-1* mRNA was successfully amplified were subsequently screened for arbovirus presence.

## Negative control reactions

In order to control for contamination in the reagents, each batch of PCR reactions contained one reaction where the cDNA template was replaced by 1 µL of sterile water.

## Polymerase chain reaction (PCR)

Samples were initially screened for the presence of Flaviviral RNA with the primer set MA/ CFD2, originally described by Kuno [21] (Table 3). PCR was performed using the GoTaq Flexi DNA polymerase system (Promega Corp., Wisconsin, USA), following manufacturer's recommendations. Briefly, each reaction contained 1.25 units GoTaq DNA Polymerase, 1X Green GoTaq Flexi Buffer, 2 mM $MgCl_2$ Solution, 0.2 mM dNTP mix, each primer at 1µM concentration and 1 µL cDNA as template. Amplification was performed by subjecting samples to 95˚C for 2 minutes, followed by 35 cycles of 95˚C for 30 seconds (denaturation), 53˚C for 30 seconds (annealing), and 72˚C for 1 minute (extension), with a final extension step of 72˚C for 5 minutes. All PCR products were visualized on 1.5% agarose gels in TBE buffer (Tris-base, Boric acid, 0.5 M EDTA at pH 8.0), containing ethidium bromide.

Samples generating positive results for the presence of *Orthoflavivirus* RNA were tested for the presence of ZIKV using a nested-PCR strategy. In a first round of amplification, we

**Table 3. Primers used for screening procedure.**

| Primer pair | Primer source | Forward primer name (and sequence) | Reverse primer name (and sequence) | Expected amplicon size |
|---|---|---|---|---|
| *MA/CFD2* | Kuno, 1998 [21] | MA (5'-CATGATGGGRAARAGRGARRAG-3') | CFD2 (5'- GTGTCCCAGCCGGCGGTGTCATCAGC-3') | 260bp |
| *ZIKENV* | Faye et al., 2008 [22] | ZIKVENVF (5'-GCTGGDGCRGACACHGGRACT-3') | ZIKVENVR (5'-RTCYACYGCCATYTGGRCTG-3') | 364bp |
| *ZIKV-NES1* | Custom-designed | ZIKV-nes1-F (5'-AAACCGTCGTCGTTTCTGGG-3') | ZIKV-nes1-R (5'-TGCATACTGCACCTCCACTG-3') | 242bp |

screened each sample with the primer set ZIKENV (Table 3), originally described by Faye et al. [22], which amplifies a 364 bp segment of the envelope protein coding region of the ZIKV genome (accession AY632535). Reagent concentrations and thermal profile were the same as previously described, excepting that annealing temperature for amplification with primer set ZIKENV was set to 55˚C.

Amplicons of the right size obtained during the first amplification were gel-purified using the Wizard SV Gel and PCR Clean-Up System (Promega Corp., Wisconsin, USA). Five μL purified amplicon were used as template for a second round of amplification using primer set ZIK-NES1 (Table 3), which was custom-designed to target a 242bp portion of the amplicon obtained with primer pair ZIKENV. Other reagent concentrations and thermal profile were the same as previously described, excepting that annealing temperature for amplification with primer set ZIK-NES1 was set to 50˚C.

### Sequencing

Amplicons of the expected size obtained after the second round of amplification were gel-purified and sent to a commercial provider (sequencing facilities at 'Universidad de las Americas' in Quito, Ecuador) for Sanger-sequencing. Sequence identity was confirmed by querying amplicon sequences against information available at the U.S. National Center for Biotechnology Information using their 'Basic Local Alignment Search Tool' (BLAST) [23].

### Infection rate (IR) estimation

IRs were calculated using the 'PooledInfRate' package, a Microsoft Office Excel add-In to compute prevalence estimates from pooled samples [24], following the software developer's recommendations. Bias-corrected maximum likelihood estimate (MLE) was chosen as point estimate because this calculation does not assume that when a pool tests positive for an arbovirus, only one individual in said pool is positive (an assumption required for the calculation of the more traditional 'minimum infection rate') [24]. All IR presented in this report correspond to values per 1,000 individuals.

### Results

A total of 6,197 mosquitoes were collected during this work. These were grouped in 2,592 pools, corresponding to 1,355 pools of female specimens, and 1,237 pools of male specimens (average pool size for each sex, region and year are shown in S1 Table). As expected, the highest abundance of both female and male pools in all four cities was observed during the rainy season (2017), followed by the post-rainy season (2018). Collections during the pre-rainy season (2016) yielded the lowest abundance of mosquito pools (Fig 2).

Only nine pools did not generate the expected results in the positive control reactions for the detection of *Actin-1* mRNA, and were therefore excluded from further analysis. The remaining 2,583 pools (corresponding to 99.6% of the total) generated the expected amplicon in the positive control reaction, indicating successful RNA extraction and cDNA synthesis. These samples were further screened for the presence of arboviral RNA.

Overall, 277 pools tested positive for ZIKV, representing 10.7% of the total number of pools screened. This corresponded to an IR of 46.9 per 1,000 individuals. The distribution of these results for each city and year is shown in Tables 4 and 5 and in Fig 3. A sub-set of the partial ZIKV sequences obtained were deposited in GenBank (www.ncbi.nlm.nih.gov/genbank/), accession numbers OM831146, OM831147, OM831148, OM831149, OM831150, OM831151, OM831152, OM831153 and OM831154.

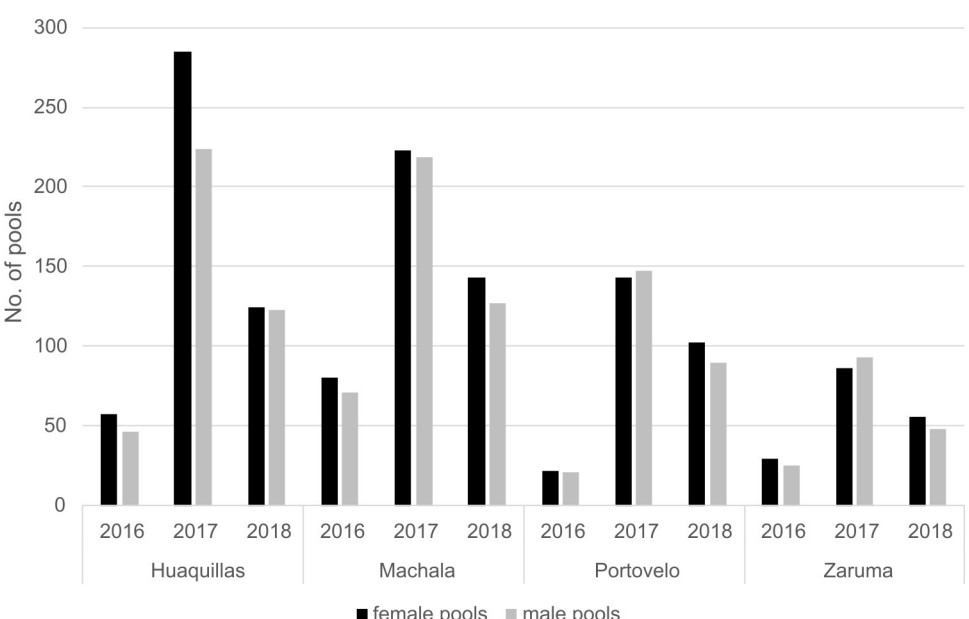

**Fig 2. Number of pools collected per city and year.** In all cities, 2017 was the year with the highest mosquito abundance. Collections were performed during the pre-rainy season in 2016, during the rainy season in 2017, and during the post-rainy season in 2018.

We detected ZIKV infection in mosquitoes of both sexes in all cities and almost all sampling periods (the only exception being male pools in the city of Portovelo during 2016). All four cities presented the highest infection rates during the 2018 season, the year in which we sampled in the post-rainy season, in both female and male pools. In fact, all cities except Zaruma showed a continuous increase in infection rates in both sexes each year throughout the sampling period.

**Table 4. Total number of pools of adult male and female *Ae. aegypti* tested (and positive for ZIKV) for each city and year.**

| City | Year | No. pools | | No. Positive | | % positive | |
|---|---|---|---|---|---|---|---|
| | | female | male | female | male | female | male |
| Huaquillas | 2016 | 57 | 46 | 1 | 2 | 1.8 | 4.3 |
| | 2017 | 285 | 224 | 25 | 14 | 8.8 | 6.3 |
| | 2018 | 124 | 123 | 24 | 17 | 19.4 | 13.8 |
| | City sub-total | 466 | 393 | 50 | 33 | 10.7 | 8.4 |
| Machala | 2016 | 80 | 71 | 4 | 2 | 5.0 | 2.8 |
| | 2017 | 223 | 219 | 23 | 10 | 10.3 | 4.6 |
| | 2018 | 143 | 127 | 50 | 25 | 35.0 | 19.7 |
| | City sub-total | 446 | 417 | 77 | 37 | 17.3 | 8.9 |
| Portovelo | 2016 | 22 | 21 | 1 | 0 | 4.5 | 0.0 |
| | 2017 | 143 | 147 | 12 | 6 | 8.4 | 4.1 |
| | 2018 | 102 | 90 | 22 | 11 | 21.6 | 12.2 |
| | City sub-total | 267 | 258 | 35 | 17 | 13.1 | 6.6 |
| Zaruma | 2016 | 29 | 25 | 3 | 1 | 10.3 | 4.0 |
| | 2017 | 86 | 93 | 4 | 1 | 4.7 | 1.1 |
| | 2018 | 56 | 48 | 11 | 8 | 19.6 | 16.7 |
| | City sub-total | 171 | 166 | 18 | 10 | 10.5 | 6.0 |

**Table 5. Infection rates (MLE per 1,000 specimens) of ZIKV in adult *Ae. aegypti* pools for each city and year.**

| City | Year | Female | | | Male | | |
|---|---|---|---|---|---|---|---|
| | | IR | 95% CI—lower limit | 95% CI—upper limit | IR | 95% CI—lower limit | 95% CI—upper limit |
| Huaquillas | 2016 | 9.4 | 0.5 | 44.6 | 18.9 | 3.5 | 59.5 |
| | 2017 | 29.9 | 20.0 | 43.0 | 20.0 | 11.5 | 32.5 |
| | 2018 | 95.2 | 64.7 | 134.5 | 73.5 | 44.9 | 113.7 |
| | Overall | 41.9 | 31.8 | 54.2 | 31.9 | 22.5 | 43.9 |
| Machala | 2016 | 27.6 | 9.1 | 64.5 | 11.2 | 2.0 | 36.1 |
| | 2017 | 45.0 | 29.6 | 65.6 | 15.2 | 7.8 | 26.9 |
| | 2018 | 240.9 | 187.4 | 302.1 | 122.1 | 81.4 | 175.8 |
| | Overall | 89.2 | 72.0 | 109.2 | 35.6 | 25.6 | 48.2 |
| Portovelo | 2016 | 35.4 | 2.1 | 157.9 | 0.0 | 0.0 | 120.5 |
| | 2017 | 43.1 | 23.7 | 71.9 | 19.1 | 7.9 | 39.3 |
| | 2018 | 99.1 | 66.6 | 141.2 | 63.1 | 33.7 | 108.1 |
| | Overall | 66.4 | 47.9 | 89.6 | 33.1 | 20.0 | 51.5 |
| Zaruma | 2016 | 67.3 | 18.5 | 168.2 | 22.7 | 1.3 | 103.7 |
| | 2017 | 18.2 | 6.0 | 42.9 | 4.2 | 0.2 | 20.3 |
| | 2018 | 94.0 | 51.5 | 156.9 | 103.2 | 48.7 | 192.0 |
| | Overall | 47.4 | 29.5 | 72.0 | 27.9 | 14.4 | 49.2 |

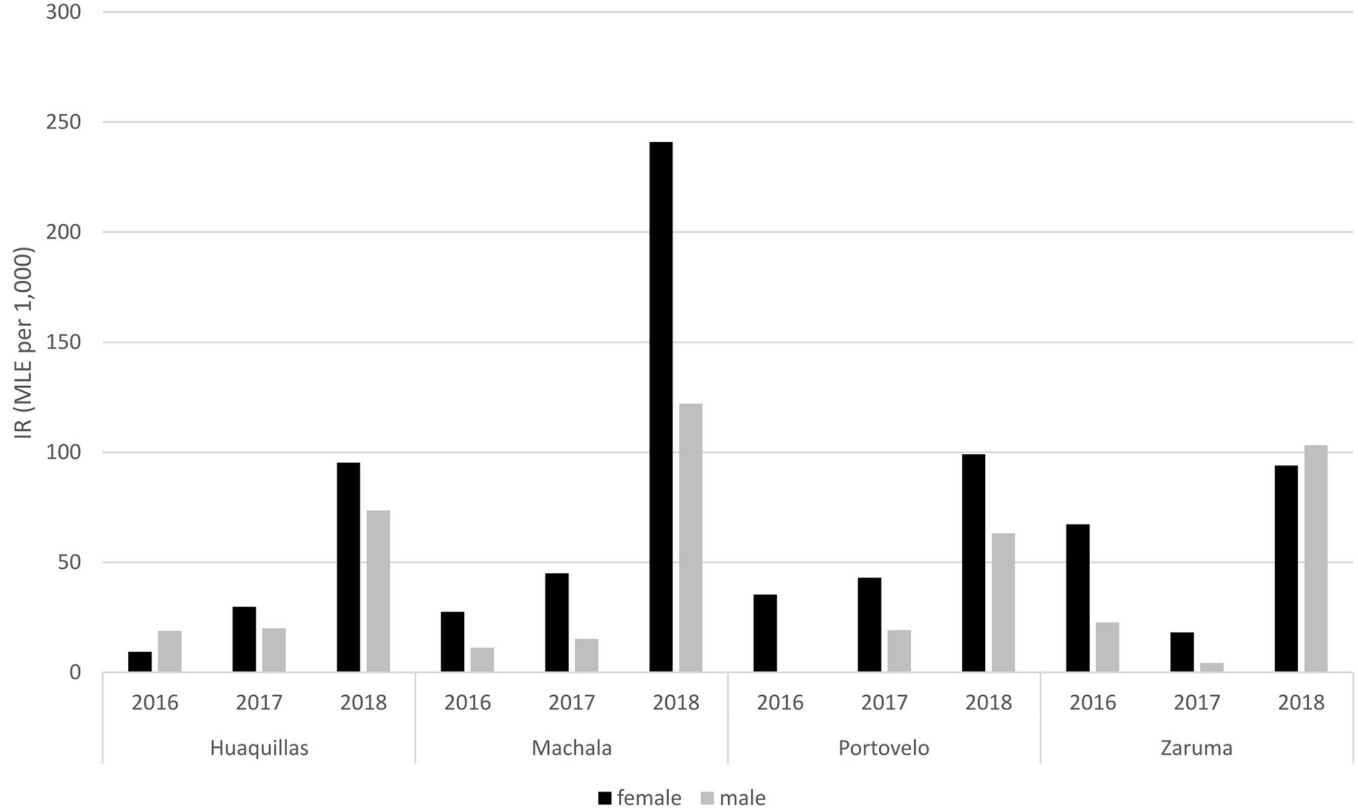

**Fig 3. Infection rates (IR) for each city and year.** For all cities, the highest IR was observed in 2018 (corresponding to the post-rainy season) in both male and female specimens.

In female mosquitoes, IRs varied over 25-fold, between 9.4 (Huaquillas, 2016) and 240.9 (Machala, 2018). Overall (i.e. all years combined) infection rates among female mosquitoes were highest in Machala (89.2), followed by Portovelo (66.4), Zaruma (47.4) and Huaquillas (41.9).

IRs among male mosquitoes were generally lower than among females, varying between 0 (Portovelo, 2016) and 122.1 (Machala, 2018). Overall infection rates in male mosquitoes were highest in Machala (35.6), followed by Portovelo (33.1), Huaquillas (31.9) and Zaruma (27.9).

## Discussion

The epidemiology of ZIKV in Ecuador remains poorly understood. After arriving in the country sometime during 2015 (as evidenced by the single human case reported for that year) (Table 1), the virus spread quickly, causing over 2,000 cases per year during 2016 and 2017, and then dropped suddenly to just 9 cases in 2018 (Fig 1) [14]. Since 2019 until the present day, only two cases have been reported in Ecuador, both in 2021 [14,25]. The epidemiological trend in neighboring countries has been similar: Colombia saw a steady decline in the number of new Zika cases, going from 91,711 in 2016 to 138 in 2022. And Peru went from 5,361 new cases in 2017, to just 20 cases in 2022 [25].

The virtual disappearance of Zika from Ecuador is in stark contrast with the epidemiological trend of dengue virus, another *Orthoflavivirus* transmitted mainly by the same vector (*Ae. aegypti*) in this country. Dengue has been consistently present in Ecuador over the last three decades [26], with a yearly incidence that, during the last five years, has ranged from approximately 3,000 cases in 2018, to over 20,000 cases in 2021 (Table 1) [14,15]. This suggests that the drastic reduction in Zika incidence observed since 2018 is not the result of a corresponding reduction in vector population, nor does it represent a reduction in the frequency of contact between vectors and human hosts. Therefore, we must look into other elements of the transmission cycle in order to shed some light into this puzzling situation.

Former reports have revealed ZIKV IRs as high as 53/1,000 [27] among female *Ae. aegypti* from provinces of the central and northern Ecuadorian coast in 2016 [27]. In agreement, our data confirms that high IRs in the *Ae. aegypti* populations were maintained throughout the study period in the southern coastal province of El Oro. Additionally, our data shows that IR in mosquitoes from all four cities progressively increased between 2016 and 2018, in spite of the fact that the number of reported Zika cases in the entire province actually declined between 2017 and 2018 (66 and 0 cases, respectively) [28]. This suggests that in our study areas, the virus is maintained by mechanisms that either do not depend on human infection, or cause extremely low rates of symptomatic infection in humans.

Although the nature of our data does not allow us to elucidate the exact nature of such mechanisms, one possibility is the maintenance of ZIKV in the mosquito populations through vertical and/or venereal transmission. Previous experimental work found high rates of ZIKV venereal transmission in *Ae. aegypti* (both male-to-female and female-to-male) [29], providing evidence that this route constitutes a mechanism for virus dissemination among the vector population. Furthermore, vertical transmission of ZIKV in *Ae. aegypti* has been reported to occur under both laboratory and natural conditions [30–32].

Because male mosquitoes can only acquire arboviral infections via either vertical or venereal route, presence of infected males in the field is generally considered as a marker of vertical and/or venereal transmission occurring in the area [33]. Interestingly, our data shows high infection rates in male as well as female mosquitoes, suggesting that these alternative transmission routes contribute to the maintenance of ZIKV in our study region. This hypothesis would be consistent with previous reports from neighboring Colombia, which found that vertical/

venereal transmission pathways play an important role for the maintenance of this virus during inter-epidemic periods [33].

Mosquitoes of the *Aedes* genus display higher rates of vertical transmission than other genera [34]. Accordingly, high rates of ZIKV vertical transmission in *Ae. aegypti* have been previously reported [35,36]. Ciota and collaborators [36] found IR as high as 28.5 among the offspring of experimentally infected female mosquitoes from South-American populations, and have stated that different *Ae. aegypti* populations might vary substantially on their ability to transmit ZIKV vertically. Although the IR among male specimens in our study could reach substantially higher values in specific populations and years, the overall (i.e. pooled for all years) IR in males for each city varies between 27.9 and 35.6 (Table 5), which is consistent with values presented in previous reports [33,36]. It is also worth noticing that due to our sampling methodology, infected mosquitoes in different pools could potentially originate from the same infected parent, thereby contributing to the relatively high IR observed in both male and female groups.

According to Reese et al. [37], wild mosquito populations in which vertical transmission has stabilized (i.e. females consistently transmit arbovirus to a high percentage of their offspring), show a high proportion of transovarially-infected individuals generating non-productive infections (i.e. infections in which no infective virions can be detected, in spite of the presence of viral antigens and nucleic acids). The study in question found that, at one of their study sites, 7 out of 58 tested specimens (equivalent to 120.7/1,000) displayed such non-productive infections. Transovarially-infected mosquitoes generating infective virions were >10 fold less abundant in these populations. If we hypothesize that a similar situation could be occurring with ZIKV and *Ae. aegypti* in our study area, this could help explain, at least in part, the low number of human infections observed during 2018 in spite of the high IR observed in our molecular screening process, as a significant proportion of mosquitoes testing positive for the presence of arboviral RNA would in fact represent non-productive infections, and would therefore play no role in the transmission of ZIKV to humans. Furthermore, this 'stabilization hypothesis' could also help explain the progressive increase in IR observed throughout our study period, as the proportion of vertically-infected specimens would be expected to increase over time. Efficient venereal transmission among adult mosquitoes could further contribute to progressive increases in IR.

To test this stabilization hypothesis, it would be necessary to evaluate the viability of viral particles obtained from field-caught mosquitoes. Unfortunately, this was not part of our original study design, and our laboratory protocol rendered all viral particles inactive at the very early steps of sample processing, making it impossible for us to quantify the proportion of productive vs non-productive infections. Therefore, we recommend that future studies in the area incorporate protocols aimed at testing the viability of virus retrieved from the vector populations. To the best of our knowledge, no further information is available regarding the prevalence of ZIKV in the mosquito populations of our study regions after 2018.

The development of herd immunity in the human population is another factor expected to contribute to the decline of ZIKV incidence in the Americas. It has been proposed that following explosive outbreaks in immunologically naive populations, large proportions of said populations can seroconvert within a relatively short time [38]. As an example, a survey conducted in northeastern Brazil found that over 60% of the local population presented serological evidence of anti-ZIKV antibody production within approximately one year of the start of the 2015 outbreak [39]. Unfortunately, no data regarding the prevalence of anti-ZIKV antibodies in the population of our study area is currently available in the scientific literature, making it difficult to estimate the extent to which herd immunity plays a role in limiting transmission in this region.

Further research is required in order to obtain a complete picture of the epidemiological landscape of ZIKV in this region. Continued monitoring of infection in field-caught mosquitoes would provide information on whether ZIKV did in fact become stabilized in these populations, while testing the viability of virus recovered from field-caught mosquitoes would help us confirm (and quantify) the existence of non-productive infections. Furthermore, performing sero-surveys aimed at identifying the prevalence of protective antibodies in the human population would allow us to identify the presence of asymptomatic infections, and estimate the protective effect of herd immunity in these areas.

To the best of our knowledge, this is the first report of vertical transmission of an arbovirus in Ecuador. Although our results cannot fully explain the almost total disappearance of clinical Zika cases in Ecuador after 2018, they provide a glimpse into the complex and dynamic ecologic interactions between arboviruses, arboviral vectors and hosts in nature.

## Supporting information

**S1 Table. Average mosquito pool size for each sex, region and year.**
(DOCX)

## Acknowledgments

The authors would like to thank Naveed Heydari for his support with sample collection.

## Author Contributions

**Conceptualization:** Anna M. Stewart-Ibarra, Sadie J. Ryan, Erin Mordecai, Marco Neira.

**Data curation:** Andrea López-Rosero, Rachel Sippy, Froilán Heras, Marco Neira.

**Formal analysis:** Andrea López-Rosero, Marco Neira.

**Funding acquisition:** Anna M. Stewart-Ibarra, Sadie J. Ryan, Erin Mordecai, Marco Neira.

**Investigation:** Andrea López-Rosero, Rachel Sippy, Anna M. Stewart-Ibarra, Froilán Heras, Marco Neira.

**Methodology:** Andrea López-Rosero, Rachel Sippy, Anna M. Stewart-Ibarra, Froilán Heras, Marco Neira.

**Project administration:** Rachel Sippy, Anna M. Stewart-Ibarra, Froilán Heras, Marco Neira.

**Resources:** Anna M. Stewart-Ibarra, Efraín Beltrán, Marco Neira.

**Supervision:** Rachel Sippy, Anna M. Stewart-Ibarra, Efraín Beltrán, Marco Neira.

**Validation:** Marco Neira.

**Writing – original draft:** Andrea López-Rosero, Marco Neira.

**Writing – review & editing:** Andrea López-Rosero, Rachel Sippy, Anna M. Stewart-Ibarra, Sadie J. Ryan, Erin Mordecai, Froilán Heras, Efraín Beltrán, Jaime A. Costales, Marco Neira.

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
