## [Decision Letter · Decision Letter 0]

27 Sep 2023

Dear Dr. Neira,

Thank you very much for submitting your manuscript "High prevalence of Zika virus infection in populations of Aedes aegypti from South-western Ecuador" for consideration at PLOS Neglected Tropical Diseases. As with all papers reviewed by the journal, your manuscript was reviewed by members of the editorial board and by several independent reviewers. In light of the reviews (below this email), we would like to invite the resubmission of a significantly-revised version that takes into account the reviewers' comments. 

We cannot make any decision about publication until we have seen the revised manuscript and your response to the reviewers' comments. Your revised manuscript is also likely to be sent to reviewers for further evaluation.

Sincerely,

Mariangela Bonizzoni

Academic Editor

Andrea Marzi

Section Editor

Reviewer's Responses to Questions

**Key Review Criteria Required for Acceptance?**

**Methods**

-Are the objectives of the study clearly articulated with a clear testable hypothesis stated?

-Is the study design appropriate to address the stated objectives?

-Is the population clearly described and appropriate for the hypothesis being tested?

-Is the sample size sufficient to ensure adequate power to address the hypothesis being tested?

-Were correct statistical analysis used to support conclusions?

-Are there concerns about ethical or regulatory requirements being met?

Reviewer #1: (No Response)

**Results**

-Does the analysis presented match the analysis plan?

-Are the results clearly and completely presented?

-Are the figures (Tables, Images) of sufficient quality for clarity?

Reviewer #1: (No Response)

**Conclusions**

-Are the conclusions supported by the data presented?

-Are the limitations of analysis clearly described?

-Do the authors discuss how these data can be helpful to advance our understanding of the topic under study?

-Is public health relevance addressed?

Reviewer #1: (No Response)

**Editorial and Data Presentation Modifications?**

Reviewer #1: (No Response)

**Summary and General Comments**

Reviewer #1: High prevalence of Zika virus infection in populations of Aedes aegypti from South-western Ecuador.

Lopez-Rosero et al. 

This is a short report describing the prevalence of Zika virus infection among the Aedes aegypti mosquitoes from four cities of El Oro province in Ecuador. The intriguing question arises: why has the Zika virus suddenly disappeared in recent years? To address this issue the authors conducted an investigation into the prevalence of Zika virus infections in mosquitoes, encompassing the peak transmission season, as well as the periods preceding and following it. 

They observed highest infection rates among mosquitoes collected during post-peak transmission season. Notably, male mosquitoes displayed significant infections, indicating evidence of vertical transmission. The authors suggested that the virus is maintained within the mosquito population via vertical transmission, particularly in the absence of reported human cases. However, authors did not delve into whether the mosquitoes harbored any infectious virus particles. The potential contribution of venereal transmission is completely ignored. The authors do not offer any insights into why mosquitoes from the post-rainy season exhibited more infections compared to those collected during the peak transmission season. A sero survey might shed light on whether asymptomatic infections or immunity resulting from natural infections have contributed to the drastic reduction in human cases. The authors should also note that several infected mosquitoes in different pools could originate from the same infected parent. 

Few minor comments:

 what is the average number of mosquitoes per pool? 

Table 3, indicate reference number under primer source

PLOS authors have the option to publish the peer review history of their article (what does this mean?). If published, this will include your full peer review and any attached files.

Reviewer #1: No
---

## [Decision Letter · Decision Letter 1]

22 Dec 2023

Dear Dr. Neira,

Thank you very much for submitting your manuscript "High prevalence of Zika virus infection in populations of Aedes aegypti from South-western Ecuador" for consideration at PLOS Neglected Tropical Diseases. As with all papers reviewed by the journal, your manuscript was reviewed by members of the editorial board and by several independent reviewers. The reviewers appreciated the attention to an important topic. Based on the reviews, we are likely to accept this manuscript for publication, providing that you modify the manuscript according to the review recommendations. 

Please consider the comment of the reviewer regarding the update on the name of the Flavivirus genus.

Sincerely,

Mariangela Bonizzoni

Academic Editor

Andrea Marzi

Section Editor

Please consider the comment of the reviewer regarding the update on the name of the Flavivirus genus.

Reviewer's Responses to Questions

**Key Review Criteria Required for Acceptance?**

**Methods**

-Are the objectives of the study clearly articulated with a clear testable hypothesis stated?

-Is the study design appropriate to address the stated objectives?

-Is the population clearly described and appropriate for the hypothesis being tested?

-Is the sample size sufficient to ensure adequate power to address the hypothesis being tested?

-Were correct statistical analysis used to support conclusions?

-Are there concerns about ethical or regulatory requirements being met?

Reviewer #1: Yes

**Results**

-Does the analysis presented match the analysis plan?

-Are the results clearly and completely presented?

-Are the figures (Tables, Images) of sufficient quality for clarity?

Reviewer #1: Yes

**Conclusions**

-Are the conclusions supported by the data presented?

-Are the limitations of analysis clearly described?

-Do the authors discuss how these data can be helpful to advance our understanding of the topic under study?

-Is public health relevance addressed?

Reviewer #1: Yes

**Editorial and Data Presentation Modifications?**

Reviewer #1: (No Response)

**Summary and General Comments**

Reviewer #1: They have addressed my concerns. However, I would like to inform the authors that the genus Flavivirus has been renamed as Orthoflavivirus.

https://ictv.global/report/chapter/flaviviridae/flaviviridae/orthoflavivirus

PLOS authors have the option to publish the peer review history of their article (what does this mean?). If published, this will include your full peer review and any attached files.

Reviewer #1: No

Figure Files:

Data Requirements:

Reproducibility:

References

---

## [Editor Report · Decision Letter 2]

8 Jan 2024

Dear Dr. Neira,

We are pleased to inform you that your manuscript 'High prevalence of Zika virus infection in populations of Aedes aegypti from South-western Ecuador' has been provisionally accepted for publication in PLOS Neglected Tropical Diseases.

Best regards,

Mariangela Bonizzoni

Academic Editor

Andrea Marzi

Section Editor

---

## [Editor Report · Acceptance letter]

15 Jan 2024

Dear Dr. Neira,

We are delighted to inform you that your manuscript, "High prevalence of Zika virus infection in populations of Aedes aegypti from South-western Ecuador," has been formally accepted for publication in PLOS Neglected Tropical Diseases.

Best regards,

Shaden Kamhawi

co-Editor-in-Chief

Paul Brindley

co-Editor-in-Chief
